# Reinforcement Learning with Elastic Time Steps

## Abstract

Reinforcement Learning (RL) is usually modelled as a Markov Decision Process (MDP), where an agent goes through time in discrete time steps. When applied outside of simulation, virtually all existing RL-based control systems maintain the MDP assumptions and use a constant rate control strategy, with a time step that is empirically chosen according to the specific application environment. Controlling dynamic systems with learned policies at the highest, worst-case frequency to guarantee stability can require high computational and energy resources, which can be hard to achieve with on-board hardware. Following the principles of reactive programming, we posit that applying control actions *only when necessary*, can allow the use of simpler hardware, reduce energy consumption, and reduce training time. To implement this reactive policy, we break the fixed frequency assumption and propose *RL with elastic time steps*, where the policy determines the next action as well as the duration of the next time step. We also derive a Soft Elastic Actor-Critic (SEAC) algorithm to compute the optimal policy in our new setting. We demonstrate the effectiveness of SEAC both theoretically and experimentally driving an agent in a simulation of simple world with Newtonian kinematics. Our experiments show higher average returns, shorter task completion times, and reduced energy consumption.

## 1 Introduction

Temporal aspects of reinforcement learning (RL), such as the duration of the execution of each action or the time needed for observations, are frequently overlooked. This oversight arises from the foundational hypothesis of the Markov Decision Process (MDP), which assumes the independence of each action undertaken by the agent (Norris, 1998). As depicted in the top section of Figure 1, conventional RL primarily focuses on training an action policy, generally neglecting the intricacies of policy implementation. Some prior researches approached the problem by splitting their control algorithm into two distinct components (Williams et al., 2017): a learning part responsible for proposing an action policy, and a control part responsible for implementing the policy (Yang et al., 2018; Zanon & Gros, 2020; Mahmood et al., 2018).

Translating action policies composed of discrete time steps into real-world applications generally means using a fixed control rate (e.g., 10 Hz). Practitioners typically choose the control rate based on their experience and the specific needs of each application, often without considering adaptability or responsiveness to changing environmental conditions. In practical applications of reinforcement learning, especially in scenarios with constrained onboard computer resources, maintaining a consistently high fixed control rate can limit the availability of computing resources for other tasks and significantly increase energy consumption.

Furthermore, in practical applications, the inherent inertia of physical systems cannot be ignored, impacting the range of feasible actions. In such cases, an agent's control actions are closely tied to factors like velocity and mass, leading to considerably different outcomes when agents execute the same actions at different control rates.

Hence, applying RL directly to real-world scenarios can be challenging when the temporal dimension is not considered. The typical approach is to employ *a fast enough but fixed control rate that accommodates the worst-case scenario* for an application (Mahmood et al., 2018), often resulting in suboptimal performance in most instances.

In this paper, we break the fixed time step assumption common in RL to create faster and more energy-efficient policies while seamlessly integrating the temporal aspect into the learning process. In our approach, the policy determines the following action and the duration of the next time step, making the entire learning process and applying policies *adaptive* to the specific demands of a given task. This paradigm shift follows the core principles of reactive programming (Bregu et al., 2016): as illustrated in the lower portion of Figure 1, in stark contrast to a strategy reliant on fixed execution times, adopting a dynamic execution time-based approach empowers the agent to achieve significant savings in terms of computational resources, energy consumption, and time expended. Moreover, our adaptive approach

enables the integration of learning and control strategies, resulting in a unified system that enhances data efficiency and simplifies the pursuit of an optimal control strategy.

An immediate benefit of our approach is that the freed computational resources can be allocated to additional tasks, such as perception and communication, broadening the scope of RL applicability in resource-constrained robots. We view elastic time steps as promising for widely adopting RL in robotics.

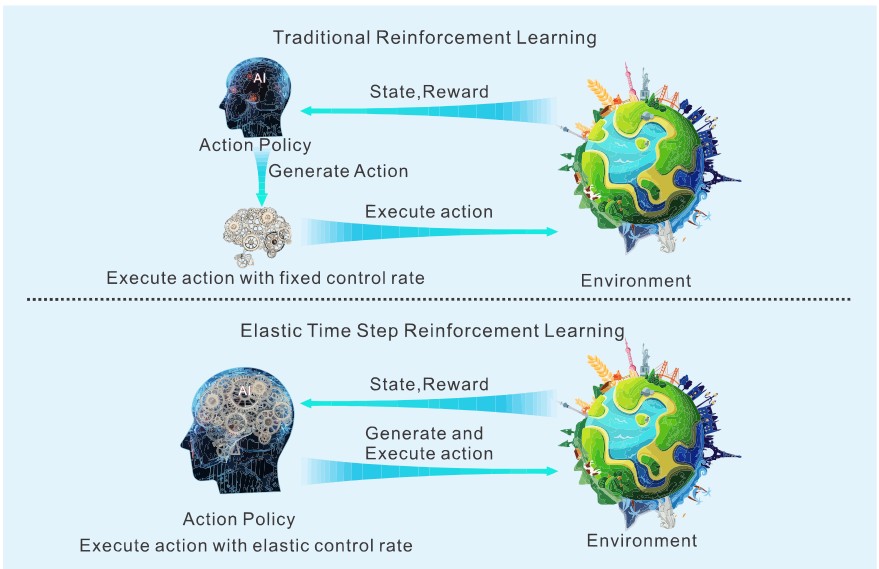

Figure 1: Comparing Elastic Time Step Reinforcement Learning and Traditional Reinforcement Learning

## 2 REINFORCEMENT LEARNING WITH A FIXED CONTROL RATE

Before delving into elastic time step-based RL, we provide a concise overview of fixed time step-based RL. A notable example of successful real-world reinforcement learning applications is Sony's autonomous racing car: Sony has effectively harnessed the synergy between reinforcement learning algorithms and a foundation of dynamic model knowledge to train AI racers that surpass human capabilities, resulting in remarkably impressive performance outcomes (Wurman et al., 2022). From a theoretical perspective, Li et al. (2020) aimed to bolster the robustness of RL within non-linear systems, substantiating their advancements through simulations in a vehicular context. A shared characteristic among these studies is their dependence on a consistent control rate, typically 10 or 60 Hz. However, it is important to note that a successful strategy does not necessarily equate to an optimal one. As previously mentioned, time is critical in determining the system's performance, whether viewed from an application or theory perspective. The energy and time costs of completing a specific task determine an agent's level of general efficiency. A superior control strategy should minimize the presence of invalid instructions and ensure control actions are executed only when necessary. Hence, the duration of an individual action step should not be rigidly fixed; instead, it should vary based on the dynamic demands of the task.

In addition to the previously mentioned scenarios, there exists a diverse range of time-sensitive reinforcement learning tasks spanning various domains. These tasks cover multiple fields, including robotics, electricity markets, and many others (Nasiriany et al., 2022; Pardo et al., 2018; Zhang et al., 2019; Yang et al., 2018). However, using a fixed control rate is a common thread among these works and systems like robots, which often lack ample computing resources, can struggle to maintain a high and fixed control rare.

## 3 REINFORCEMENT LEARNING WITH ELASTIC TIME STEPS

A straightforward approach to variable time step duration is to monitor the completion of each execution action and dispatching the subsequent command. Indeed, in low-frequency control scenarios, there are typically no extensive demands for information delay or the overall time required to accomplish the task (games like Go or Chess, Silver et al., 2016; 2018). However, in applications like robotics or autonomous driving, the required control frequency can

vary from very high (Hwangbo et al., 2017; Hester & Stone, 2013; Hester et al., 2012) to low depending on the state of the system. Following reactive programming principles(Bregu et al., 2016), to control the system only when necessary we propose that the policy also outputs the duration of the current time step. Reducing the overall number of time steps conserves computational resources, reduces the agent's energy consumption, and enhances data efficiency.

Unfortunately, in most RL algorithms, such as Q learning (Watkins & Dayan, 1992) and the policy gradient algorithm (Sutton et al., 1999a), there is no concept of the action execution time, which is considered only in few works (Ramstedt & Pal, 2019; Bouteiller et al., 2021). When control frequency is taken into consideration, it is mostly related to specific control problems (Adam et al., 2011; Almási et al., 2020), and assuming actions executed at a fixed rate.

We propose a reward policy incorporating the agent's energy consumption and the time taken to complete a task and extend Soft Actor-Critic Haarnoja et al. (2018a) into the Soft Elastic Actor-Critic (SEAC) algorithm, detailed in the following.

It is worth noting that our current implementation uses a partial Model-Predictive Control system and omits some components that would be necessary for a real-world implementation, e.g. a proportional-integral-derivative controller (PID) controller (Singh et al., 2013), an Extended Kalman filter (EKF) (Dai et al., 2019), and other essential elements. These components would need to be implemented to use SEAC into a real system environment. Nevertheless, we show that our system can indeed learn the duration of control steps and outperform established methods in a proof-of-concept implementation.

### 3.1 MULTI-OBJECTIVE REWARD POLICY

As shown in Figure 2, our approach tackles a multi-objective optimization challenge, in contrast to conventional single-objective reinforcement learning reward strategies. We aim to achieve a predefined objective (metric 1) while minimizing energy consumption (metric 2) and time to complete the task (metric 3). To reduce the reward to a scalar, we introduce 3 weighting factors: $\alpha_t$, $\alpha_\varepsilon$, and $\alpha_\tau$, respectively assigned to our three metrics. It is important to note that we consider only the energy consumption associated with the computation of a time step (i.e. energy is linearly proportional to the number of steps) and not the energy consumption of the action itself (e.g moving a heavy object, taking a picture, etc.). Thus, our assessment of the agent's energy usage is solely based on the *computational load*.

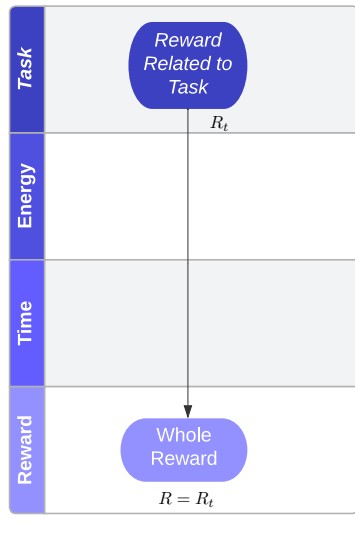
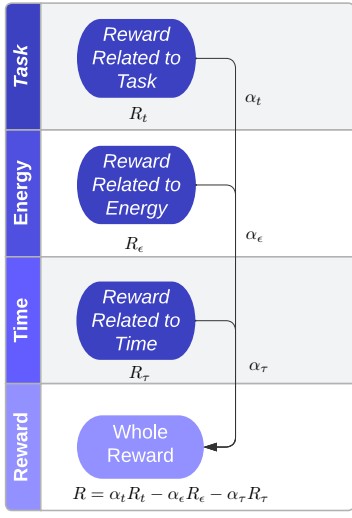

(a) Traditional RL  (b) Elastic Time Step RL

Figure 2: (a) the reward policy for traditional RL; (b) the reward policy for elastic time step RL

We assume that each action incurs a uniform energy consumption, denoted as $\varepsilon$, and the total number of steps required to accomplish a task is n. Consequently, the total energy consumed to complete a task is $n \cdot \varepsilon$. Similarly, the time taken

to execute an action is $\tau$, and the overall time required is $n \cdot \tau$. In this context, the aggregate reward for task completion is represented by r, and the relationship can be expressed as follows:

**Reward Function**

**Definition 1** *The reward function is defined as:*

$$R = \alpha_t \cdot R_t - \alpha_\varepsilon \cdot R_\varepsilon - \alpha_\tau \cdot R_\tau \qquad (1)$$

*where $R_t = n \cdot r$, $R_\varepsilon = n \cdot \varepsilon$ and $R_\tau = n \cdot \tau$, with n the total number of time steps, $\tau$ the time taken to execute a time step, $\varepsilon$ the energy cost of a time step, and $\alpha_t, \alpha_\varepsilon, \alpha_\tau$ being parametric weighting factors. We determine the optimal policy $\pi^*$, which maximizes the reward R.*

We validate our reward strategy by updating the SAC algorithm and implementing fully connected neural networks (Müller et al., 1995) as both the actor and critic strategy. We assume the agent can explore the unknown environment as much as possible based on information entropy, giving a high probability that the agent can discover the optimal solution to complete the task.

In contrast to the conventional RL, we incorporate additional inputs in the form of states at the network's input layer, including the time spent performing the previous action ($T_{t-1}$), and the actual distance moved in the previous step ($M_{t-1}$). Additionally, our approach involves an extra component at the output: the execution time for each action. As shown in Figure 3, we formally define the structure of SEAC. $Q_t$ means the Q value, $Log\_A_t$ means the distribution parameters of action values. $A_t^{predict}$ is the predict value of Actor policy, used to compute the loss function (Definition 3). $\alpha$ means the influence factor of information entropy on the Bellman equation (Haarnoja et al., 2018a).

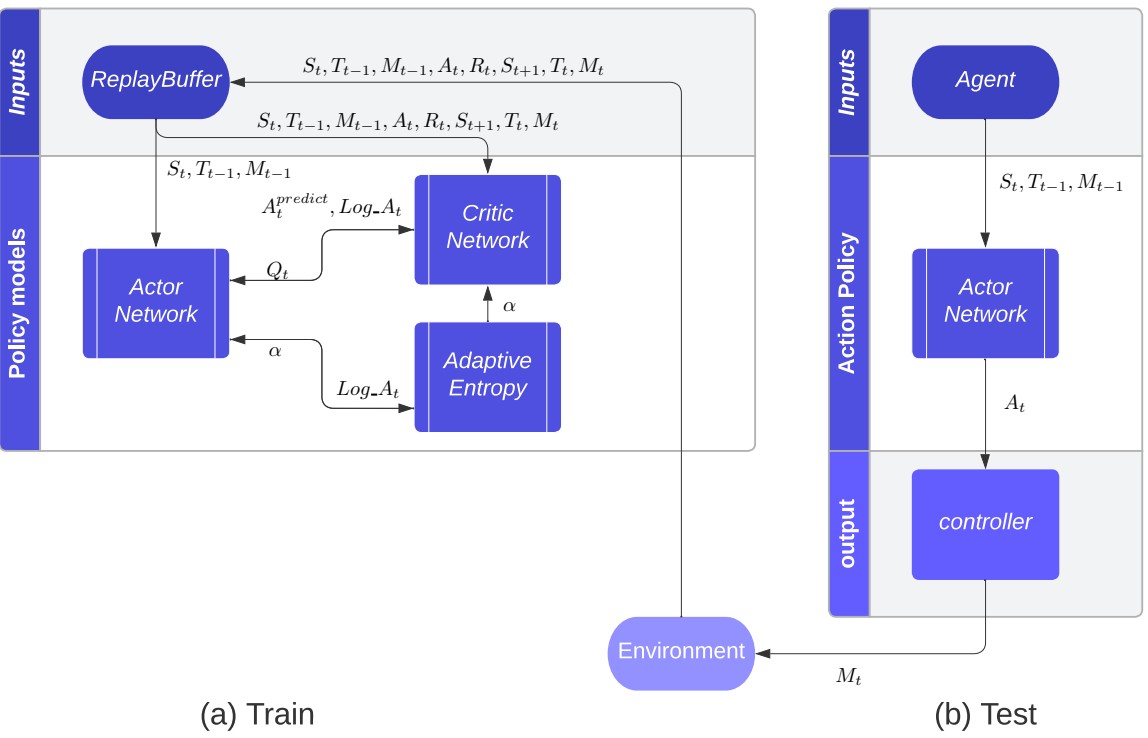

(a) Train                         (b) Test

Figure 3: (a) the train part of SEAC; (b) the test part of SEAC

Unlike traditional RL, our approach involves collecting not only the state information ($S_t$), action value ($A_t$), and reward value ($R_t$) but also the actual impact values of the execution ($M_{t-1}$) and action duration ($T_{t-1}$) from the previous time step. Under the context of our test environment, we define the movement value component of $A_t$ as the target movement distance. At the same time, $M_t$ represents the actual distance moved, considering the effects of inertia and friction. This supplementary information is essential as it aids the neural network in learning the correct

action execution time and action value within the current environment. Therefore, we include these variables alongside the state information in the neural network input.

When the Actor Network generates the action value $A_t$ for the next step, the controller (Figure 3) will computes a range of control-related parameters (e.g., speed, acceleration, etc. under the context of our test environment) based on the action value and time. Ultimately, the agent incorporates these actionable parameters into the environment, generating a new state and reward. This process is iterated until the completion of the task.

Our objective is for the agent to learn the optimal execution time for each step independently. We need to ensure that time is not considered as a negative value. Consequently, diverging from the single $Tanh$ (Kalman & Kwasny, 1992) output layer typical in traditional RL Actor Networks, we separate the Actor Network's output layer into two segments: we use $Tanh$ as the output activation for the action value, and $ReLu6$ (Howard et al., 2017) for the output activation related to the time value.

## 3.2 ENVIRONMENT DESIGN

Our SEAC architecture requires additional input and output information that is not available within existing RL environments, we establish a test environment based on Gymnasium featuring variable action execution times, shown in Figure 4:

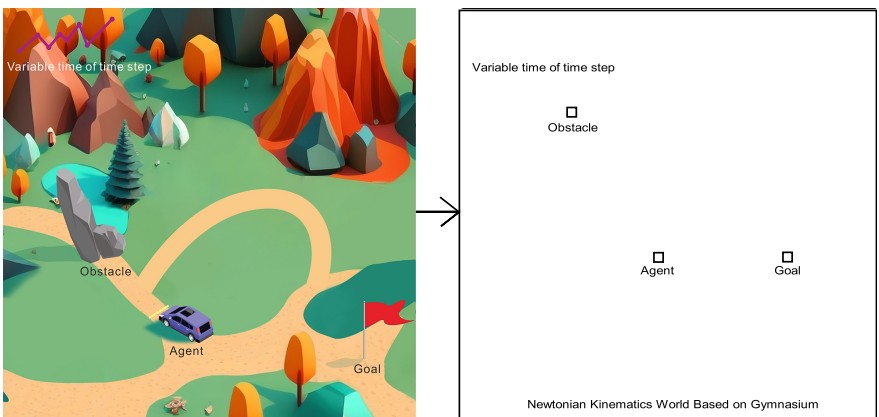

Figure 4: A simple Newtonian Kinematics environment designed for verifying SEAC based on gymnasium.

This environment is a continuous two-dimensional (2D) and consists of a starting point, a goal, and an obstacle. The task involves guiding an agent from the starting point to the goal while avoiding the obstacle. Upon resetting the environment, a new goal and obstacle are randomly generated. The conclusion of an epoch is reached when the agent reaches the goal or encounters an obstacle. The agent is governed by Newton's laws of motion, including friction.

The starting point of the agent is also randomly determined. If the goal or obstacle happen to be too close to the starting point, they are reset. Similarly, if the goal is too close to the location where the obstacle was generated, the obstacle's position is reset. This process continues until all three points are situated at least 0.05 meters apart from each other on a (2 x 2) meters map. Meanwhile, the maximum moving distance for a single step is 0.1 meters.

There are six dimensions of the state in the environment: the agent's position, the position of the obstacle, the position of the goal, the velocity of the agent, and duration of the preceding time step. It is worth noting that we are indeed using historical data (i.e. the duration of the preceding step), but but refrain from using recurrent neural networks (Zaremba et al., 2014; Lipton et al., 2015, RNNs, ). This decision stems from our concern that adopting recurrent architectures might deviate the overall reinforcement learning process from the Markov assumption (Norris, 1998; Gers et al., 2000): different decisions could arise from the same state due to the dynamic environment. While this scenario might not entirely align with the Markov assumption, it works as a Semi-MDP (Sutton et al., 1999b). For a comprehensive understanding of the semi-Markov process setup within our environment, please refer to Appendix A.

We consider 3 dimensions to the actions within the environment:

1. The time taken by the agent to execute the action;
2. the expected movement distance of the agent along the x axis;
3. the expected movement distance of the agent along the y axis.

For instance, an action $a_t = (0.2, 0.1, -0.1)$ denotes that the agent is expected to move 0.1 meters along the x axis and -0.1 meters along the y axis within 0.2 seconds. For more detailed environment settings, see Appendix B.

## 4   POLICY SET AND IMPROVEMENT

Like SAC, SEAC makes use of the entropy-augmented soft value function. Definition 2 comes from Haarnoja et al. (2018a), the Bellman equation can also be estimated by augmenting the reward function with an entropy reward. If we consider $T_t$ and $M_t$ as parts of $S_t$, then:

**Entropy concerned Bellman Equation**

**Definition 2** *The policy starts from any function* Q: S × A → ℝ, *and repeatedly applies a modified Bellman backup operator* $\tau^\pi$ *given by:*

$$\tau^\pi Q(s_t, a_t) \triangleq r(s_t, a_t) + \gamma \mathbb{E}_{s_{t+1} \sim p}[V(s_{t+1})] \tag{2}$$

*where*

$$V(s_t) = \mathbb{E}_{a_t \sim \pi}[Q(s_t, a_t) - \log \pi(a_t \mid s_t)] \tag{3}$$

Our primary focus is validating the reward policy associated with elastic time steps and assessing the impact of adaptive action execution times on the reinforcement learning algorithm. Consequently, we have refrained from altering the loss function of the Critic Network:

**Loss Function of the Critic Network**

**Definition 3** *The SEAC critic loss is:*

$$L_v^{SEAC}(\psi) = \mathbb{E}_{s_t \sim D}[\frac{1}{2}(V_\psi - \mathbb{E}_{a_t \sim \pi_\phi}[Q_\theta(s_t, a_t) - \log \pi_\phi(a_t \mid s_t)])^2] \tag{4}$$

Based on the same consideration, the loss function of the Actor Network is also consistent with the loss function of SAC:

**Loss Function of the Actor Network**

**Definition 4** *The SEAC actor loss is:*

$$L_\pi^{SEAC}(\pi) = \mathbb{E}_{(s_t, a_t) \sim D}[\frac{1}{2}(Q_\theta(s_t, a_t) - \hat{Q}(s_t, a_t))^2] \tag{5}$$

*with:*

$$\hat{Q}(s_t, a_t) = r(s_t, a_t) + \gamma \mathbb{E}_{s_{t+1} \sim p}[V_{\bar{\psi}}(s_{t+1})] \tag{6}$$

As Definition 1, the precise reward configuration for our environment are outlined in Table 1. The hyperparameters settings can be found in Appendix C.

Table 1: Reward Policy for The Simple Newtonian Kinematics Environment

| | Reward Policy | |
|---|---|---|
| Name | Value | Annotation |
| | 25.0 | Reach the goal |
| r | $-25.0$ | Crash on an obstacle |
| | $-1.0 \cdot D_{goal}$ | $D_{goal}$: distance to goal |
| $\epsilon$ | 1.0 | Computational energy (Joule) |
| $\alpha_t$ | 1.0 | Task gain factor |
| $\alpha_\epsilon$ | 1.0 | Energy gain factor |
| $\alpha_\tau$ | 1.0 | Time gain factor |

# 5 EXPERIMENTAL RESULTS

We conducted eleven experiments for each of the three RL algorithms, employing various parameters within the environment described in subsection 3.2 [1]. These experiments were conducted on a machine equipped with an Intel Core i7-10700K CPU and an NVIDIA RTX 2080 GPU, running Ubuntu 20.04. Subsequently, we selected the best-performing policy for each of these three algorithms to draw the graphs in Figures 5–8.

The frequency range for action execution spans from 1 to 100 Hz, and the agent's speed value ranges from -2 to 2 meters per second. We compared our results with the original SAC (Haarnoja et al., 2018b) and PPO (Schulman et al., 2017) algorithms, both employing a fixed action execution frequency of 5.0 Hz.

We use the conventional average return graph and record the average time cost per task to provide a clear and intuitive representation of our approach's performance. Furthermore, we generate a graph illustrating the variation in action execution frequency for six epochs with the SEAC model. Finally, we employ a raincloud graph to visualize the disparities in energy costs among these three RL algorithms for one hundred missions.

Appendix C provides all hyperparameter settings and implementation details. The average return results of all algorithms are shown in Figure 5, and their time-consuming results are shown in Figure 6:

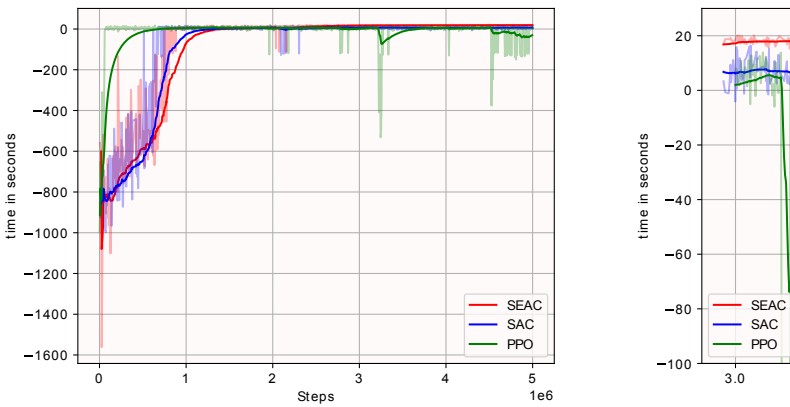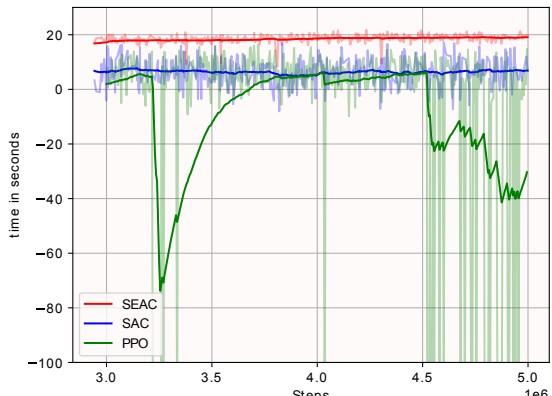

Figure 5: Average returns for three algorithms trained in five million steps.

Figure 5 and Figure 6 show that SEAC surpasses the baselines in terms of average return and time efficiency. PPO, an on-policy algorithm that does not consider information entropy, exhibits quicker convergence during training when compared to SAC and SEAC. However, its final performance displays more significant fluctuations. In contrast, SEAC, using the same policy optimization algorithm but incorporating an elastic time step, demonstrates higher and more stable final performance than SAC.

When considering the adaptation of action execution frequency within the SEAC model, we generate frequency diagrams for six distinct trials, as depicted in Figure 7, each utilizing different random seeds. Additionally, Figure 8 illustrates the energy cost (i.e. the number of time steps) across one hundred independent trial: as expected, SEAC minimizes energy with respect to PPO and SAC without affecting the overall average reward. It is worth noting that SAC and PPO are not optimising for energy consumption, so they are expected to have a large result spread. More interestingly, SEAC *both* reduces energy consumption *and* achieved a high reward. We maintain a uniform seed for all algorithms during this analysis to ensure fair and consistent results.

As shown in Figure 7, the agent's task execution strategy primarily focuses on minimizing the number of steps and the time required to complete the task. Notably, the agent often invests a substantial but justifiable amount of time in the initial movement phase, followed by smaller times for subsequent steps to arrive at the goal. This pattern aligns with our core philosophy of minimizing energy and time consumption.

---

[1]Our code is publicly available, we will add link after blind peer review ends

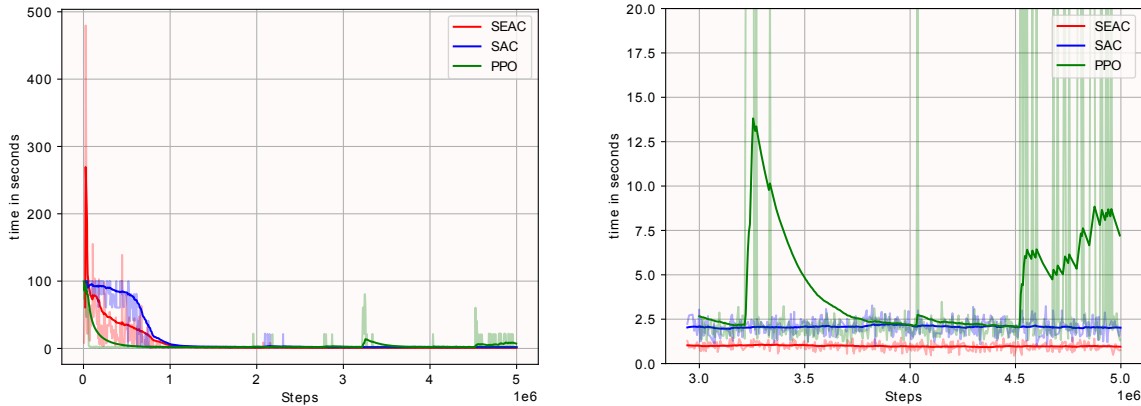

Figure 6: Average time cost per epoch for three algorithms trained in five millions steps.

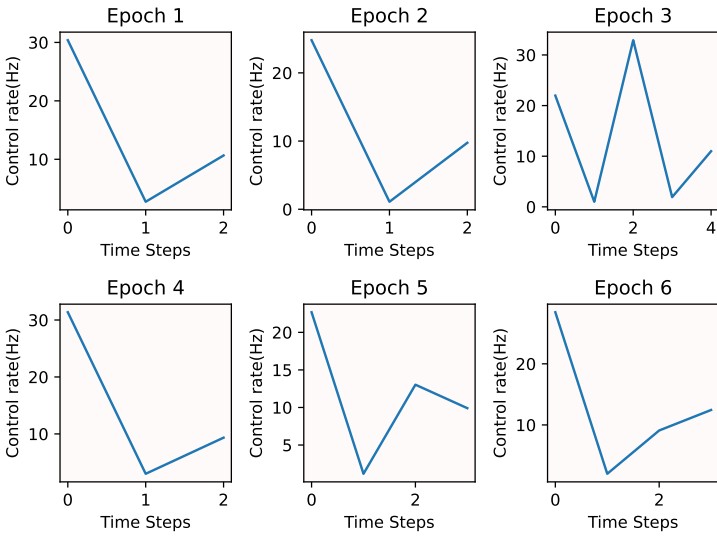

Figure 7: Six example configurations that show how SEAC dynamically changes the control rate to adapt to the task for different time steps.

Furthermore, the variance in data distribution is notably reduced within the SEAC results. These findings underscore the algorithm's heightened stability in dynamic environments, further substantiating the practicality of our elastic time step-based reward policy.

## 6 CONCLUSIONS AND FUTURE WORK

We propose an elastic time step-based reward policy that allows an agent to decide the duration of a time step in reinforcement learning, reducing energy consumption and increasing sample efficiency (since fewer time steps are needed to reach a goal). Reducing the number of time steps can be very beneficial when using robots with limited capabilities, as the newly freed computational resources can be used for other tasks such as perception, communication, or mapping. The overall energy reduction also increases the general sustainability of robotics missions.

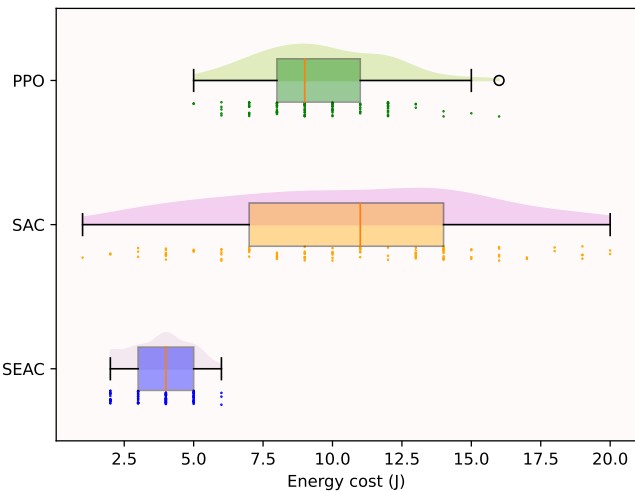

Figure 8: Energy cost for 100 trials. SEAC consistently reduces the number of time steps compared with PPO and SAC without affecting the overall average reward. SAC and PPO are not optimising for energy consumption and have therefore a much larger spread.

We introduce the Soft Elastic Actor Critic (SEAC) algorithm and verify its applicability with a proof-of-concept implementation in an environment with Newtonian kinematics. The algorithm could be easily extended to real-world applications, and we invite the reader to refer to section 5 and Appendix C for the implementation details.

To the best of our knowledge, SEAC is the first reinforcement learning algorithm that simultaneously outputs actions and the duration of the following time step. Although the method would benefit from testing in more realistic and dynamic settings, such as Mujoco(Todorov et al., 2012) or TMRL (tmrl, 2023), we believe this method represents a promising approach to make RL more efficient.

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

# A A SEMI-MARKOV DECISION PROCESSING SETTING WITH OUR DYNAMIC ENVIRONMENT

For reinforcement learning that considers time step lengths, we have one assumption:

**Strategic uniqueness and independence**

**Hypothesis 1** *The generation strategies $\pi_A$(generate the action) and $\pi_V$(provide the speed) are unique and independent. In this paper, the stratgies can be written as:*

$$A_{t+1} = \pi_A(S_t) \tag{7}$$

$$V_{t+1} = \pi_A(S_t) \times \pi_A(S_{t-1}) \tag{8}$$

Because $\pi_V$ uses the history values of $S_{t-1}$, and reward function R can be determined as:

$$R_t = E(S_t|A_t), R_t \in \mathbb{R} \tag{9}$$

Since it is a "end of strike" process, the history value from t to $\chi$ could be summarized as:

$$H_{t-\chi} = (S_t, A_t, R_t, S_{t+1}, A_{t+1}, R_{t+1}, S_{t+2}, A_{t+2}, R_{t+2}...S_\chi, A_\chi, R_\chi) \tag{10}$$

The set of all history values can be marked as a set $\omega$, Semi-MDP options can be defined as:

**Options**

**Definition 5** *Options*
*For this Semi-MDP, the Options can be written as:*

$$O = (\zeta, \pi_\tau, \beta) \tag{11}$$

*with:*
$\zeta \in S$ *is the initial state set for this Option.*
$\pi_\tau$ *is the policy can be summarized as $\pi_\tau = \pi_A \cap \pi_V$.*
$\beta \to [0,1]$ *represents the probability of termination of the Option in a specific state. Since $\pi_\tau$ would not be an empty set. Therefore, $\beta \neq 0$*
$\pi_\tau : \omega \times A \to [0,1]$.
$\beta : \omega \to [0,1]$

For this Semi-MDP, we have:

**V values distribution**

**Definition 6** *V values distribution:*

$$V^{\pi_A}(s) = \mathbb{E}\left\{\sum_{k=1}^{\chi-t} \gamma^{k-1} r_{t+k} | \varepsilon(\pi_\tau, s, t)\right\} \tag{12}$$

*with:*
$\varepsilon(\pi_\tau, s, t)$ *means on t time step and state s, agent follows poliy $\pi_\tau$*

**policy $\pi_\mu$**

**Definition 7** *Options strategy $\pi_\mu : S \times O \to [0,1]$: Return the probability of selecting the next option according to the current state, until the option exits, and then proceed to the next round of selection.*

$$Q^{\pi_\mu}(s,o) = \mathbb{E}\left\{\sum_{k=1}^{\chi-t} \gamma^{k-1} r_{t+k} | \varepsilon(o\pi_\mu, s, t)\right\} \tag{13}$$

*with:*
$\varepsilon(\pi_A, s, t)$ *represents First act according to option: o, after exiting, select the next option according to $\pi_\mu$.*

Based on these three definitions and one hypothesis, this Semi-MDP can be summarised as follows:

**Semi-MDP settings**

**Definition 8** *Semi-MDP settings*
*Rewards:*

$$r_s^o = \mathbb{E}\left\{r_{t+1} + \gamma r_{t+2} + ... + \gamma^{k-1} r_{t+k} | \varepsilon(o, s, t)\right\} \tag{14}$$

*The environment and motion model:*

$$p_{ss'}^o = \sum_{k=1}^{\infty} p(s', k)\gamma^k \tag{15}$$

*Bellman equation under policy $\pi_\mu$:*

$$V^{\pi_\mu}(s) = \mathbb{E}\left\{r_{t+1} + \gamma r_{t+2} + ... + \gamma^{k-1} r_{t+k} + V^{\pi_\mu} s_{t+k} | \varepsilon(\pi_\mu, s, t)\right\}$$

$$= \sum_{o \in O_s} \pi_\mu(s, o) \left[r_s^o + \sum_{s' \in S} p_{ss'}^o V^{\pi_\mu}(s')\right] \tag{16}$$

*with:*

$$Q^{\pi_\mu}(s, o) = r_s^o + \sum_{s' \in S} p_{ss'}^o \sum_{o' \in O_s} \pi_\mu(s', o') Q^{\pi_\mu}(s', o') \tag{17}$$

*In the optimal case, we have:*

$$V_O^*(s) = \max_{o \in O_s} \mathbb{E}\left\{r_s^o + \sum_{s' \in S} p_{ss'}^o V_O^*(S')\right\}$$

$$= \max_{o \in O_s} \mathbb{E}\left\{r + \gamma_k V_O^*(s') | \varepsilon(o, s)\right\} \tag{18}$$

$$Q_O^*(s, o) = \mathbb{E}\left\{r_{t+1} + \gamma r_{t+2} + ... + \gamma^{k-1} r_{t+k} + \gamma_k \max_{o' \in O_{s_{t+k}}} Q_O^*(s_{t+k}, o') | \varepsilon(o, s, t)\right\}$$

$$= r_s^o + \sum_{s'} p_{ss'}^o \max_{o' \in O_{t+k}} Q_O^*(s', o') \tag{19}$$

$$= \mathbb{E}\left\{r + \gamma_k \max_{o' \in O_{t+k}} Q_O^*(s', o') | \varepsilon(o, s)\right\}$$

Under this definiton, to find $V_O^*$ means get the optimal policy $\pi_\mu^*$.

# B VALIDATION ENVIRONMENT DETAILS

The Spatial Information of our environment are:

Table 2: Details of The Simple Newtonian Kinematics Gymnasium Environment

| Environment details | | |
|---|---|---|
| Name | Value | Annotation |
| Action dimension | 3 | |
| Range of speed | $[-2, 2]$ | |
| Action Space | $[-0.1, 0.1]$ | |
| Range of time | 1.0 | |
| State dimension | 6 | Task gain factor |
| World size | $(2.0, 2.0)$ | in meters |
| Obstacle shape | Round | Radius: 5cm |
| Agent weight | 20 | in $Kg$ |
| Gravity fcator | 9.80665 | in $m/s^2$ |
| Static friction coeddicient | 0.009 | |

The state dimensions are:

> **State**
>
> **Definition 9** *State*
> *We have only one starting point, endpoint, and obstacle in the set environment. The positions of the endpoint and obstacle are randomized in each epoch.*
>
> $$S_t = (Pos, Obs, Goal) \tag{20}$$
>
> *Where:*
> *Pos = Position Data of The Agent in X and Y Direction,*
> *Obs = Position Data of The Obstacle in X and Y Direction,*
> *Goal = Position Data of The Goal in X and Y Direction.*

The action dimensions are:

> **Action**
>
> **Definition 10** *Action*
> *Each action should have a corresponding execution duration.*
>
> $$a_t = (a_t, a_{mx}, a_{my}) \tag{21}$$
>
> *Where:*
> $a_t$ *= The time of The Agent to implement current action,*
> $a_{mx}$ *= The Moving Target Point of The Agent in X Coordinate,*
> $a_{my}$ *= The Moving Target Point of The Agent in Y Coordinate.*

## C  HYPERPARAMETER SETTING OF SEAC

Table 3: Hyperparameters Setting of SEAC

| Hyperparameter sheet | | |
|---|---|---|
| Name | Value | Annotation |
| Total steps | $5e6$ | |
| $\gamma$ | 0.99 | Discount factor |
| Net shape | $(256, 256)$ | |
| batch_size | 256 | |
| a_lr | $3e5$ | Learning rate of Actor Network |
| c_lr | $3e5$ | Learning rate of Critic Network |
| max_steps | 500 | Maximum steps for one epoch |
| $\alpha$ | 0.12 | |
| $\eta$ | $-3$ | Refer to SAC (Haarnoja et al., 2018b) |
| $T_i$ | 5.0 | Time duration of rest compared RL algorithms, in HZ |
| min_frequency | 1.0 | Minimum control frequency, in HZ |
| max_frequency | 100.0 | Maximum control frequency, in HZ |
| Optimizer | Adam | Refer to Adam (Kingma & Ba, 2014) |
| environment steps | 1 | |
| Replaybuffer size | $1e6$ | |
| Number of samples before training start | $5 \cdot max\_steps$ | |
| Number of critics | 2 | |

