# OpenReview forum: "Reinforcement Learning with Elastic Time Steps"
_ICLR.cc/2024/Conference — Submitted to ICLR 2024_

### Official Review · Reviewer_FcT3 · 2023-10-20

**Soundness:** 2 fair
**Presentation:** 2 fair
**Contribution:** 1 poor
**Rating:** 3
**Confidence:** 4

**Summary:**

The main contribution of this paper is the idea that, in RL, a policy can be made to specify both a control action to apply *and* the length of time an actuator should apply that action. The paper integrates this idea within an existing, popular algorithm for model-free RL (the SAC algorithm), and presents comparative results in a small example problem.

**Strengths:**

As far as I know this precise idea is novel, and it is certainly intuitive. Results and other details aside, I think the community should investigate this direction more deeply and this paper provides a nice starting point for that effort.

**Weaknesses:**

Weaknesses:
- The literature review is quite thin, and as a circumspect reader I do wonder how novel this idea really is, given how little literature is referenced. For example, a quick google scholar search reveals the following papers that seem to be very closely related: [1, 2]. I would also add that variable rate decision making is widely studied in the control theory literature. A key phrase to find this literature is “adaptive time step.”
- The paragraph immediately above section 3.1 indicates that there are a lot of loose ends that are not being discussed in detail, and which may strongly affect results. The imprecision of this discussion (e.g., what is a “partial MPC” and what role does the PID serve if you already use MPC?) suggests that the work may be somewhat immature.
- The reward structure discussed section 3.1 is *not* what one would properly call a “multi-objective optimization problem.” A distinguishing trait of such problems is the concept of “Pareto optimality” which encodes all of the tradeoffs among optimal performance with respect to each separate objective. By assuming a fixed weighting, this paper effectively reduces the problem to a standard optimization problem (and picks a single point on the Pareto frontier). I recommend consulting [3] for further details.
- Relatedly, the construction in Definition 1 is not as clear as it could be. For instance: are the R terms intended to be functions of state (and action)? If so, why does it make sense to only accrue reward at the times when actions are changed? Doesn’t that lead to some obvious opportunities for reward hacking? For example, could an agent decide to plow straight through some region of low reward for a bunch of (unactuated) time steps? Also, can R_t and R_\epsilon be evaluated at every time, or only at the end of an episode? Evidently, at every time t, but then I am lost as to why the agent is incentivized to minimize n, the length (in steps) of an elastic time step. I am lost.
- The details of the method are really not very clearly explained. For example, throughout the discussion of section 3 it appears that the there is some notion of an agent physically moving and the policy gets to access a measure of distance somewhere. This is unclear: everything up until this point (and in general) is framed around general MDPs, which have nothing to do with physical embodiment. How general-purpose is the proposed approach?
- Relatedly, the test environment is not very clearly explained, or at the very least, suggests a very basic question: wouldn’t it make more sense for the policy to output a force, rather than a target position? This would remove the need for lower-level tracking control (MPC, PID) and also mitigate the “measure of distance” question above, I believe.
- I do not follow the “six dimensions of the state in the environment” - in fact, I count 9: 2 each for agent/obstacle/goal position, 2 for agent velocity, and 1 for duration. What am I missing? In the same paragraph, the discussion of semi-Markov processes and recurrence is rather opaque. Use of words like “might” and “could” lead me to wonder how clearly this point is understood. I suggest clarifying the language here.
- There are no discernible error bards in the plots, and the shaded areas appear to be traces of other plotted data - this needs to be explained precisely, and plots should show some measure of error in order to be interpreted statistically.
- More importantly, even: there is little to no interpretation of the behavior of the proposed policies. Results here indicate some differences in aggregate behavior (although the interpretation to that effect should really require some error bars as above), but it would really help to understand what is going on if the authors expanded upon Fig. 7 to illustrate what was going on in the environments in these situations and why it made sense to change the control rate as shown.

Other nitpicks:
- It seems like the main motivation here is one of saving computational resources. Obviously, most control systems are pretty lightweight and so I imagine these savings really come in from the perception side, e.g., if you no longer have to process big images at high frame rate. Experimental results to illustrate these savings more directly than the abstraction of “number of repeated actions” would be highly motivating.
- There are quite a few typos and other small syntax issues.
- The vertical axis labels are wrong in Fig. 5.
- Figures 5 and 6 could be more clear about indicating that the right hand sides are insets of the left. Also, why were the methods run for so long - it seems they all converged quite a bit earlier and then for some reason PPO destabilized. Something seems off here.
- Why does Fig. 7 say “epochs” instead of “configurations?”


[1] Chen, Y., Wu, H., Liang, Y., & Lai, G. (2021, July). VarLenMARL: A framework of variable-length time-step multi-agent reinforcement learning for cooperative charging in sensor networks. In 2021 18th Annual IEEE International Conference on Sensing, Communication, and Networking (SECON) (pp. 1-9). IEEE.

[2] Sharma, Sahil, Aravind S. Lakshminarayanan, and Balaraman Ravindran. "Learning to Repeat: Fine Grained Action Repetition for Deep Reinforcement Learning." International Conference on Learning Representations. 2016.

[3] Deb, Kalyanmoy, and Kalyanmoy Deb. "Multi-objective optimization." Search methodologies: Introductory tutorials in optimization and decision support techniques. Boston, MA: Springer US, 2013. 403-449.

**Questions:**

Please see my comments above.

---

> ### Author Response · Authors · 2023-11-22
>
> Thank you for acknowledging the merits of our paper. We share your conviction
> that the elastic time step method holds significant potential, particularly in
> scenarios with constrained computational resources in real-world applications. We
> are exploring further developments in this direction.
>
> At the same time, we extend our thanks for bringing to light the deficiencies in
> our paper. We are committed to addressing each of them.
>
> - In response to Q1, Thanks to the two references. We spent some time reading
>   them and agreed that our literature review section does need further
>   refinement. The core of these two articles is standby action or repeated
>   action to achieve changes in control rate, and they are all based on fixed
>   frequency without exception. They do not directly pursue the goal of
>   minimizing the number of control actions. We will include
>   these two articles in our literature review and clarify our differences.
>
> - In response to Q2, as described in Section 3.1, our model only provides the
>   execution time of each action, which is only one part of the control of an
>   actual robot. We cite MPC as just one of many possible control algorithms. We
>   agree that the description can cause confusion, and we will modify the
>   text to "an appropriate control algorithm."
>
> - In response to Q3, it is a pertinent suggestion. Our core idea is to free as
>   many computing resources as possible, which can then be used for perception,
>   security, etc. Our description here can be improved according to the Definition
>   1, we scalarize this multi-objective problem into a single value through
>   weighting. We will remove the description of "multiple-objective" to ensure
>   there is no ambiguity.
>
> - In response to Q4, yes, we expect R to be a function of state (and action).
>   The reward R_t for the task can be arbitrarily defined, and it does not
>   necessarily require an action change to obtain the reward value. We will
>   improve our description of Definition 1 to underline that R is used to
>   evaluate each step individually.
>
> - In response to Q5, Due to space limitations, we have put the measurements
>   of the agent and environment in Appendix B. We have only
>   done experiments in this simple kinematic environment so far. The conversion
>   formulas are：
>
> D_{aim} = 1/2 \cdot (V_{aim}+V_{current}) \cdot T;
> V_{aim} = V_{current} + AT;
> F_{aim} = mA;
> F_{true} = F_{aim} – f_{friction};
> f_{friction}=\mu mg, if F_{aim} > f_{friction}, else: f_{friction} = F_{aim}.
>
> D_{aim} is the distance that the agent needs to move as computed by the policy,
> T is the time to complete the movement as generated by the policy, m is the mass
> of the agent, \mu is the friction coefficient, and g is the gravity acceleration.
> Through the movement distance and time generated by the strategy, we can know
> the target speed required to complete the strategy and then calculate the
> acceleration. Knowing the acceleration, we can find the force. However, due to
> friction, the actual speed generated will be inconsistent with the target speed,
> and the movement of the agent will be affected by Newtonian kinematics. Our
> method aims at minimizing the use of computing resources by minimizing the
> number of steps and time required to complete the task.
>
> - In response to Q6, it is a pertinent suggestion. Setting the output to force
>   is indeed a better choice. We will add additional experiments to the paper.
>
> - In response to Q7, yes, you are right. We initially thought that time and past
>   action values should not belong to the state. But we agree they should also be
>   correctly defined as part of the state. We will restate the relevant
>   definitions.
>
> - In response to Q8, based on your feedback, we will conduct multiple
>   experiments (for example, six sets) and use this data to redraw Figures 5 and
>   6.
>
> - In response to Q9, Figure 7 records the time-related generation strategy of
>   the SEAC model in this environment. We do not have additional drawings because
>   SAC and PPO are both fixed times (such as 5 Hz mentioned in the article).
>   Figure 7 needs to be examined together with Figure 8: the SEAC model strategy
>   shows the use of fewer time steps in Figure 8. In Figure 6, SEAC also has a
>   shorter completion time, showing an advantage of SEAC in terms of total energy
>   consumption. We will add some additional explanation diagrams based on Figure
>   7 to explain the working of SEAC, SAC, and PPO strategies to make our work
>   more understandable and readable.
>
> - In response to Q10, thank you for recognizing our core concept of reducing the use of
>   computing resources. Based on your feedback, we will focus more on this aspect
>   in the text if modifications are needed. Unfortunately, we have not
>   yet completed the hardware experiment.
>
> - In response to Q11, thank you for pointing out grammar issues and typos; we will
>   try to improve the text as much as possible.
>
> - In response to Q12, thank you for pointing this out. We will fix the axis
>   of Figure 5.

---

> ### Author Response · Authors · 2023-11-22
>
> - In response to Q13, because the environment and tasks we set are relatively
>   simple, these algorithms have converged around 3 million steps. However, we
>   noticed that PPO would be unstable in this final period. We are not fully able to
>   explain the behavior of the neural network, and we are reporting it as is.
>
> - In response to Q14, thanks for the feedback. It should not be "epoch" but
>   "episode." We will fix it.

---

> > ### Comment · Reviewer_FcT3 · 2023-11-22
> >
> > Thank you for your response to my comments.

---

### Official Review · Reviewer_DZcS · 2023-10-28

**Soundness:** 2 fair
**Presentation:** 3 good
**Contribution:** 1 poor
**Rating:** 3
**Confidence:** 4

**Summary:**

This work presents relaxes the fixed frequency assumption of MDP typically studied in RL and proposes RL with elastic time steps.  Also a Soft Elastic Actor-Critic algorithm is derived with theoretical and practical benefits.

**Strengths:**

1. The work is concisely summarized.
2. The use of elastic time is important in the tasks such as robotics etc.

**Weaknesses:**

1. There are many existing studies with varying time (e.g. option framework, action repetitions…)
Authors introduce some notions of options and semi-MDP in appendix, but without clear definitions of each notation, which makes it harder to see the clear connections to the main work and the option framework.  (It was not clear how the authors validated Bellman-like equations for elastic time case)  Assuming the algorithm is properly derived from the option framework, it is necessary to compare to the existing work based on the framework.  (Or at least it should show significant practical results compared to the existing work; it seems the experiments are not for sufficiently complex tasks.)
2. Existing environments such as OpenAI Gym can be easily adjusted to include time as information for states; I am not sure what the authors mean by “...additional input and output information that is not available within existing RL environments…”
(Note that simulators anyway need to run with small time interval to maintain accuracy, and action durations can be just a repetition of that.)
4.  Figure 5 is a bit hard to parse: why time in seconds are negative?  I could guess this but it is better to make them crystal clear.
5.  It would be better to show baseline with 100Hz (fixed) case, not 5.0 Hz since the elastic one uses 1 to 100 Hz.
6.  Figure 7 is also hard to interpret; why are there only 2 time steps…?  2 steps are enough to complete tasks…?
7.  Finally, it was not clear why the authors specifically used the reward defined in Definition 1.

**Questions:**

1.  Figure 4 right seems too sparse; what does it try to imply?
2.  What is the action space A?  Is it the Cartesian product of “action” and “time”?

---

> ### Author Response · Authors · 2023-11-22
>
> Thank you for your appreciation of our work. Our core idea is to reduce the
> computational resources the agent requires by minimizing the number of steps
> (each of which requires to generate an action) rather than simply generating the
> execution time of the action variable. For example, in simulation, there may be
> no difference between executing an action of 1 second 10 times or executing an
> action once for 10 seconds. However, their computational cost differs greatly in
> a physical, embedded environment. A computer engineering analysis would be
> comparing polling a resource as opposed to using an interrupt signal: in the former
> case, the computer constantly checks if a resource is available, while in the latter
> the computer is notified when the resource is available and can access it only
> when necessary.
>
> - In response to Q1, our work differs from action repetition or the options
>   framework (HRL for extended actions). These methods do not reduce the amount
>   of steps and thus do not reduce computational resources. For example, the
>   agent with SEAC policy can reduce the frequency at which it samples the
>   environment. However, we will add relevant references to
>   improve our literature review so that readers can more clearly understand the
>   differences with our work.
>
>   As for the issue in Appendix A, we will modify the relevant description to
>   clarify it. Although we do not have complex experimental results, we are
>   working in this direction. Our core idea can save computing resources compared
>   with methods such as action repetition.
>
> - In response to Q2, additional input refers to the historical value of the
>   agent, the duration of the previous step, and the action value of the last
>   step. The other output refers to the duration of the action to be performed.
>   We have a maximum frequency limit to ensure accurate execution of the
>   action, which is 100Hz. The above information can be found in Sections 3 and 5
>   of the article.
>
> - In response to Q3, thanks for pointing it out. We will fix Figure 5.
>
> - In response to Q4, it is a good proposal. We can do more groups, such as 1 Hz,
>   10 Hz, 60 Hz, 100 Hz, etc. We will use your feedback to improve our
>   experiments.
>
> - In response to Q5, our environmental tasks are relatively simple. Please refer
>   to Appendix B for specific physical information. Figure 7 needs to be viewed
>   together with Figure 8: to complete a task, the average number of steps
>   required by SEAC is 3, and the average number of steps needed by SAC is more
>   than 10. This is a reflection of our success in reducing computational resources
>   and time. In addition, we will add some model explanation diagrams based on
>   Figure 7.
>
> - In response to Q6, Figure 4 is a situation where we simplify an autonomous
>   driving problem into a simple Newtonian mechanics environment. Figure 4 just
>   shows the environment that we use to verify our algorithms. Please refer to
>   Appendix B for specific physical information. We will adjust Figure 4 so it
>   does not look so empty.
>
> - In response to Q7, the action set is (time for next step, move distance on X,
>   move distance on Y). Yes, it is the Cartesian product of "time" and "action".
>   The specific information can be viewed in Definition 10 of Appendix B.

---

> > ### Comment · Reviewer_DZcS · 2023-11-22
> > **Thank you for your response**
> >
> > Thank you for your responses;
> > I still have concerns that the existing work could be easily used to reduce the amount of steps if the reward/state are properly designed.
> > If the focus of this work is really about reducing the required steps, I would like to see more focused arguments around it.
> > Although I see some potential benefits of this work in the future, I keep my score for now.
> > But thank you for your clarification; it makes me feel that the work should become more solid in the future.

---

> > > ### Author Response · Authors · 2023-11-23
> > > **Additional comment on the number of steps**
> > >
> > > We would like to stress the fact that the objective is not simply reducing the number of steps, but rather using the minimal number of steps to achieve a desired performance. Imagine driving a car in a straight line in a very large environment without obstacles: the number of control actions needed are very few. On the contrary, driving in tight spaces with low tolerances and following complex paths can require a higher control frequency. Without elastic time steps the control frequency is set to a fixed value which must be the worst case scenario that allows controllability of the system (or an average value if safety is not a concern) at all times. By allowing the system to learn its control frequency according to environmental conditions we can reduce its computational and energy cost as well as allow finer control when needed.
> > > We are currently testing our method on a more realistic scenario using TrackMania (TMRL) and we can reduce the number of time steps around 20% when switching the control frequency from 1 to 100 Hz, whereas a standard RL system would work at a fixed 20Hz.

---

### Official Review · Reviewer_XuXs · 2023-10-30

**Soundness:** 2 fair
**Presentation:** 2 fair
**Contribution:** 2 fair
**Rating:** 6
**Confidence:** 3

**Summary:**

The paper proposes a reactive reinforcement learning policy, which breaks the fixed time step assumption commonly adopted in RL and determines the next action and the duration of the next time step as input to the controller, thus integrating the temporal aspect into the learning process. The authors test their approach in a simulation of a simple word with Newtonian kinematics, showing its effectiveness in leading to higher efficiency in terms of speed and energy consumption.

**Strengths:**

The contribution is clearly stated and it is relevant to the development of real-world efficient and effective RL-based control systems. The paper structure is well organized and clear. Figures and schemes are helpful and explanatory. Limitations of the proposed approach (which components would be necessary for a real-world implementation) are clearly stated.

**Weaknesses:**

The contribution is relevant but it is limited compared to the existing state of the art. Since the contribution is mainly aimed to applying RL control outside of simulation, a proof of concept of the functioning of the proposed algorithm on a real-world application (rather than only in a simulation environment) would be important, in my view.
Although the paper’s quality of presentation is generally fair, I found the comparison with the related works poor and lacking of an insightful discussion about existing time-sensitive RL tasks, which are only quickly listed at the end of section 2. Expanding such a paragraph could make the relevance and applicability of the paper’s contribution clearer.
The presentation of the results could also be improved (see specific comments on the next section).

**Questions:**

-	Fig 1: I don’t find Fig 1 completely effective, based on the description within the Introduction. Since one of the contributions of the Elastic Time Step RL is that of enabling the policy to output the time step duration, together with the action, this could be somehow explicitly indicated in the Figure. Also, even though I understand the intention of splitting the “learning” and “execution” part of a RL implementation, I find the brain-like icon confusing when used to indicate the “execution” rather than the “learning” component of the system.
-	I would be curious to know from which specific practical application (robotics, autonomous driving?) comes the authors’ inspiration for the paper.
-	Page 4, sentence preceding Definition 1: “The aggregate reward for task completion is represented by r”. Did you mean “R” (capital letter)?
-	The paragraph after Definition 1 (“We validate our reward strategy…”) could be rephrased to highlight SEAC differences compared to SAC.
-	What do you mean when you say “…giving a high probability that the agent can discover the optimal solution to complete the task”?  Maybe this sentence can be rephrased to make the exploration strategy clearer.
-	In general, from the sentence starting “we assume the agent…” to the sentence ending with “…Bellman equation”, I find the flow of the text, which can be read while referring to the scheme on Fig 3, a little hard to follow, in the sense that it jumps from one block to another one (of the Fig.3) without a precise order. Incorporating more references to the visual scheme and aligning the text with the functional flow of the figure 3 (rather than simply listing the meaning of the symbols) could help the readability.
-	You mention that one major contribution of the SEAC is to include the execution time of each action to the output, but this term is not explicitly indicated on Fig.3, together with the At.
-	The meaning of the double arrows in Fig.3 is not very clear to me. Maybe an explanation could be included either on the caption or on the main text.
-	The impact value of the execution is defined, based on the chosen environment, as the target movement distance. Do you have in mind some examples of different implementations for different problems?
-	In the end of paragraph 3.1, when you say “the controller will compute a range of control-related parameters”, is this represented by Mt?
-	In the end of paragraph 3.1, when you say “our objective is for the agent to learn the optimal execution time”, is the execution time equivalent to the action time, and therefore represented by Tt?
-	Typo: “but but” in the sentence starting with “it is worth noting…” in paragraph 3.2
-	What is the meaning of “p” in eq. (2)?
-	Since the SAEC loss functions are (if I understand well) equal to those of SAC, rather than simply reporting the definitions, I would suggest to reorganize Section 4 to better explain how your formulation of the reward function is included in the update steps of the RL algorithm.
-	Section 5: When you refer to the “three RL algorithms”, do you mean SEAC, SAC and PPO? In this case, you should first say that you are comparing SEAC results with SAC and PPO in the text, otherwise it is not clear to the reader.
-	What are you representing differently on the left and right side of Fig. 5 and 6? Is it the right side simply a y-axis zoom-in of the left side? You should specify it on the figures' captions. What is the legend for the lighter colored plots?
-	I think that Fig. 7, as it is, is not very informative. It shows that SEAC dynamically changes the control rate, but it doesn’t allow to evaluate whether it does it in a meaningful way. Showing the scenario and/or information about the corresponding actions would make the concept clearer.
-	I feel Fig.8 would be more readable by inverting x and y axes (evaluation metric on the y-axis). Furthermore, you mention the overall reward both in the section and in the figure caption, but is the overall reward shown somewhere?

---

> ### Author Response · Authors · 2023-11-22
>
> Thank you for your comments. We are dedicated to enriching our literature
> review to give readers a comprehensive understanding of how our algorithms are
> applied in real-life scenarios.
>
> - In response to Q1, it is a pertinent suggestion. SEAC itself does not contain
>   an execution unit. It only provides the action value required for execution
>   and the time of the action. We can optimize it by replacing the brain icon with
>   a clock and changing the description of the arrow to "Execute an action with a
>   fixed amount of time."
>
> - In response to Q2, onboard computing can be severely constrained in many
>   robots. It is worth mentioning that the core of article [1] is the opposite of
>   ours: [1]1 aims to control as fast as possible, while we strive to control as
>   little as possible.
>
> - In response to Q3, yes, thanks for your correction. We will fix it.
>
> - In response to Q4, it is a pertinent suggestion. The loss function of SEAC is
>   consistent with that of SAC. As you proposed next, we will modify the
>   expression here so that people can clearly understand that we only implement
>   Definition 1 through SAC. The same idea can be achieved with other types of
>   reinforcement learning algorithms. It is worth noting that we didn't create a
>   new reinforcement learning algorithm.
>
> - In response to Q5, we will rewrite it as "make the agent learn the best time and
>   corresponding action policy".
>
> - In response to Q6, Q7, and Q8, they are pertinent suggestions. We can modify
>   Figure 3 and its caption describe more clearly how some low-level
>   parameters, such as time t, etc., participate in the training process.
>
> - In response to Q9, yes, such as the classic robot arm problem: one assumes
>   that the output of the strategy set is a force set. Suppose you want to
>   quickly grab a fragile object, such as an egg, without considering other
>   interferences, such as the material of the robot arm. In that case, the grip
>   strength of the robot arm and direction are essential, and the duration of the
>   force is even more critical. Any application of reinforcement learning
>   algorithms in real-world environments of continuous control type can adopt our
>   elastic time step idea.
>
> - In response to Q10, no, M_t refers to the movement value of the previous step.
>   The physics-related properties of the vehicle itself do not, and should not,
>   participate in network training. Please refer to Appendix B. The series of
>   related control parameters here refer to speed and acceleration.
>   As we mentioned in another reply, we will modify Figure 3 and the
>   corresponding marks to make our work more readable and understandable.
>
> - In response to Q11, yes, the execution time equals the action time.
>
> - In response to Q12, sorry, we will fix it.
>
> - In response to Q13, p is the last step. We will improve the expression of
>   Definition 2.
>
> - In response to Q14, it is a good suggestion. We can rephrase Section 4 to show
>   how some low-level parameters work with the model training based on a modified
>   Figure 3.
>
> - In response to Q15, it is a good suggestion. We will modify the description to
>   explain what kind of algorithms we are comparing.
>
> - In response to Q16, our original intention with Figure 7 was to inform readers
>   that SEAC takes fewer time steps to achieve the set goals. The agent will
>   first take a long step to achieve the goal and then fine-tune it. We will add
>   some model explanation diagrams based on Figure 7 to explain the workings of
>   SEAC, SAC, and PPO strategies to make our work more understandable and
>   readable.
>
> - In response to Q17, it is a good suggestion. We can change it to a rain cloud
>   image. Figure 5 is realistic about the overall reward. As defined in 1, the
>   total reward includes the number of steps and time used.
>
> [1] Bouteiller, Yann, et al. "Reinforcement learning with random delays." International conference on learning representations. 2020.
>
> [2] Bregu, Endri, et al. "Reactive control of autonomous drones." Proceedings of the 14th Annual International Conference on Mobile Systems, Applications, and Services. 2016.

---

### Official Review · Reviewer_VwVw · 2023-10-30

**Soundness:** 2 fair
**Presentation:** 2 fair
**Contribution:** 1 poor
**Rating:** 3
**Confidence:** 4

**Summary:**

This paper extends the classical RL setting, where there is no concept of the action execution time, to RL with elastic time steps. The authors propose SEAC to output the next action as well as the duration of the next time step.

**Strengths:**

The proposed problem is interesting. The figures are vivid, and the paper is easy to follow.

**Weaknesses:**

The contribution and novelty is vague. As for the traditional RL, the control frequency is only an abstract definition. I think the proposed framework can be seen as a special instance of the traditional RL framework given a reformulated action space / state space / reward function. The algorithm also seems quite like SAC with new state / actions. Also, what is the relationship between the proposed algorithm with HRL methods?

**Questions:**

See above.

---

> ### Author Response · Authors · 2023-11-22
>
> Thank you for your comments. The framework we propose should be a flexible time-step
> algorithm that is versatile and minimizes energy loss and time. SEAC is
> implemented on top of SAC, but this idea can be applied to other off-policy
> algorithms. Please imagine you do not need high-frequency computing of
> environmental perception-related components, such as images, point clouds, and
> additional information: the freed computational resources can be used to make
> robots cheaper or more reactive to the environment.
>
>
> In response to your question. HRL, as we understand it, aims to solve the sparse
> reward problem. When the environmental rewards are too light, the agent may be
> unable to obtain samples with positive rewards for a long time, which brings
> difficulties in learning value functions and strategies. Therefore, it has a
> high-level reward policy and at least one low-level reward policy. In our case,
> we have only one policy. What’s more, the focus of HRL is not to reduce the
> amount of calculation, nor does it help reduce the time. One can use HRL to
> learn the duration of time steps to plan a large goal and multiple subgoals
> and set different rewards, but this will require more complex models to be
> trained, increasing the computational burden on the agent.
>
> In short, our algorithm is not an HRL algorithm. Its ideas could be implemented
> in an HRL framework, but our goals are not the same.
>
> We will add HRL and repeat related references to improve our literature review
> so that readers can more clearly understand the differences between our and
> option algorithms.

---

### Meta-Review · Area_Chair_2Kon · 2023-12-09

**Metareview:**

This paper focuses on the issue of choosing decision frequency in a real-world setting. Instead of taking actions at a fixed frequency, this work proposes to take them when necessary to reduce computational and energy costs. The approach of RL with elastic time is proposed where the next action, as well as the duration time, is generated by the agent.


Strengths:

- The work addresses an interesting problem.

Weaknesses:

- Reviewers brought up the question of how it relates to hierarchical RL (HRL). Although the authors mention that HRL has a different goal, the more pertinent point here is that this work falls within the umbrella of temporally extended actions. Hence, a connection to HRL cannot be avoided, and yet such connections are ignored in the work and the rebuttal. Some definitions regarding options are given in the appendix, with no clear connection made to them from the main text.

- Related work that addresses similar issues is scantly covered. Specifically, the work by [Karami et al. (2023)](https://arxiv.org/abs/2212.04407) is very similar to the proposed approach.

- For a work focused on a real-world setting, a real-world environment such as with robots would have substantially compensated for the lack of theoretical insights.

- There are several flaws with the experimental setup, such as utilizing existing Gym environments, comparison with option framework and action repetition, and having high fixed frequencies for actions as the baseline, pointed out by the reviewers that will result in a new set of experiments and reviews.

Overall, the reviewers gave excellent feedback. I highly recommend the authors incorporate them in their resubmission to a future venue. However, the current paper clearly does not meet the expectations of this venue.

**Justification For Why Not Higher Score:**

There are many issues with the paper, as I list in the weaknesses above. The work is not as thorough or rigorous as one would expect for this venue, and based on the responses, the authors do not seem to be aware of this gap.

**Justification For Why Not Lower Score:**

N/A

---

### Decision · Program_Chairs · 2024-01-16

Reject